# Genome-Wide Identification of the NRT1 Family Members and Their Expression under Low-Nitrate Conditions in Chinese Cabbage (*Brassica rapa* L. ssp. *pekinensis*)

**DOI:** 10.3390/plants12223882

**Published:** 2023-11-17

**Authors:** Yihui Zhang, Faujiah Nurhasanah Ritonga, Shu Zhang, Fengde Wang, Jingjuan Li, Jianwei Gao

**Affiliations:** 1Shandong Branch of National Vegetable Improvement Center, Institute of Vegetables, Shandong Academy of Agricultural Science, Jinan 250100, China; zyh_0923@163.com (Y.Z.); ritongafaujiah@ymail.com (F.N.R.); shuzhang2013@126.com (S.Z.); wfengde@163.com (F.W.); 2Graduate School, Padjadjaran University, Bandung 40132, West Java, Indonesia

**Keywords:** NRT1, Chinese cabbage, gene structure, genotype, nitrogen

## Abstract

Nitrate transporters (NRTs) actively take up and transform nitrate (N) to form a large family with many members and distinct functions in plant growth and development. However, few studies have identified them in the context of low nitrate concentrations in Chinese cabbage (*Brassica rapa* L. ssp. *Pekinensis*), an important vegetable in China. This study focuses on the identification and analysis of the nitrate transporter 1 (NRT1) gene family as well as various aspects, including its phylogenic distribution, chromosomal position, gene structure, conserved motifs, and duplication pattern. Using bioinformatics methods, we identified and analyzed 84 *BrNRT1* genes distributed on ten chromosomes. Furthermore, we conducted an analysis of the expression profile of the NRT1 gene in various tissues of Chinese cabbage exposed to varying nitrate concentrations. A phylogenetic analysis revealed that *BrNRT1s* members are distributed in six distinct groups. Based on an analysis of gene structure and conserved motifs, it can be inferred that *BrNRT1* exhibits a generally conserved structural pattern. The promoters of *BrNRT1* were discovered to contain moosefs (MFS) elements, suggesting their potential role in the regulation of NO_3_^−^ transport across the cell membrane in Chinese cabbage. A transcriptome study and a subsequent RT-qPCR analysis revealed that the expression patterns of some *BrNRT1* genes were distinct to specific tissues. This observation implies these genes may contribute to nitrate uptake and transport in various tissues or organs. The results offer fundamental insights into investigating the *NRT1* gene family in Chinese cabbage. These results provide basic information for future research on the functional characterization of *NRT1* genes in Chinese cabbage and the elucidation of the molecular mechanisms underlying low nitrogen tolerance in Chinese cabbage.

## 1. Introduction

Nitrogen (N) is a crucial element due to its function in leaf and growth development, seed germination and dormancy, floral induction, and response to stress [1,2,3]. Nitrate (NO_3_^−^) is a significant source of N for higher plants and a signal that regulates plant development [4,5,6]. Nitrate–nitrogen is the primary source of inorganic N for plants in soils with a high oxygen content. The spatial and temporal variations in the nitrate–nitrogen concentration result from leaching and microbial activity. However, plants have developed physiological and morphological adaptations to cope with a fluctuating soil nitrate supply. In addition, it was found that the regulatory networks of the transport proteins involved in nitrate uptake are complex in plants.

The significant development of nitrate transporter families was identified and characterized at the molecular level in recent years. However, the absorption and transportation of nitrate in plants primarily depend on the involvement of the NRT1, nitrate transporter 2 (NRT2) [7], and chloride channel (CLC) multigenic families [8,9]. NRT1 is a member of the vast PTR (peptide transporter) family, which includes members known as proton-coupled oligopeptide transporter (POT) and solute carrier 15 (SLC15) [7,10]. NRT1 was initially considered to be a low-affinity nitrate transporter. Several recent studies found that NRT1 is a dual-affinity transporter, such as *AtNRT1.1* in *A. thaliana* and *MtNRT1.3* in *Medicago truncatula* [11,12]. In addition, *ZmNRT1.1B* was found to be a high-affinity nitrate transporter in *Zea mays* [13]. Recently, the NRT1 nitrate transporter family was studied extensively due to its specificity in plant growth and development. Since the first discovery of NPF6.3/AtNRT1.1/CHL1 proteins in *Arabidopsis thaliana* [14], additional NRT1.1 homologs have been cloned and functionally classified in various plant species [15,16]. Currently, 53, 93, and 91 *NRT1* family members have been identified from *Arabidopsis*, rice, and sorghum, respectively [17,18,19]. In Arabidopsis, NRT1 members have been demonstrated to play an important role as transporters of nitrate and other substrates [20,21]. *NRT1* genes are expressed in roots, leaves, and xylem sap. *AtNRT1.1* and *AtNRT1.2* are important for absorbing nitrate from the soil solution into root cells [22,23]. Furthermore, *AtNRT1.9* plays a crucial role in transporting nitrates from root stelae to root phloem to improve downward nitrate transport in roots [24]. *NRT1.4* contributes to nitrate storage in leaf petioles [25]. *NRT1.5* and *NRT1.8* are important for loading and recovering nitrate from xylem sap [26,27]. *NRT1.6* is involved in the transfer of nitrate from maternal tissue to developing embryos [28]. Nitrate is transferred from older to younger leaves through *NRT1.7*, *NRT1.11*, and *NRT1.12* [29,30].

Chinese cabbage, scientifically known as *Brassica campestris* L. ssp. *Chinensis* L., a species in the family Brassicaceae, is known to have originated in China. At present, Chinese cabbage, known as *Brassica rapa* L. ssp. *Pekinensis*, is an important vegetable crop in China due to its prominent position in terms of its cultivation area inside the country. In addition, Chinese cabbage plays a crucial role in ensuring a stable supply of vegetables and maintaining market equilibrium in China [31]. As Chinese cabbage is a leafy vegetable, it accumulates nitrates during its growth and development [32]. To enhance the quality and productivity of Chinese cabbage, the N intake in Chinese cabbage fields is often 2–5 times greater than the recommended level for Chinese cabbage cultivation [33]. A previous study confirmed that 12 kg of pure N per an area of 667 m^2^ is recommended without the concurrent use of organic fertilizer [33]. Therefore, research on the molecular mechanism of N uptake in Chinese cabbage might contribute to addressing these issues, as well as provide theoretical support for the study of NRT1.1 in other species. However, the research on nitrate transporter genes and their impact on overall efficiency in Chinese cabbage are limited compared to studies conducted on Arabidopsis, rice (*Oryza sativa* L.), and tomato (*Solanum lycopersicum*).

In this study, we identified and analyzed 84 *BrNRT1* genes and several of their aspects, including their phylogenic distribution, chromosomal position, gene structure, conserved motifs, and duplication pattern under low-nitrate (LN) conditions in the Chinese cabbage genome. Furthermore, we conducted an analysis of the expression profile of the NRT1 gene in various tissues of Chinese cabbage exposed to varying nitrate concentrations. Our findings are expected to serve as a genetic base and foundation for comprehending the involvement of the NRT1 genes in the process of nitrogen uptake in Chinese cabbage.

## 2. Results

### 2.1. Identification of BrNRT1s in Chinese Cabbage

We identified 84 protein sequences of *BrNRT1* using all AtNRT1 amino acid (aa) sequences from the Chinese cabbage genome, and their basic characteristics are listed in Appendix A. The 84 *BrNRT1* genes were distributed over ten chromosomes. The lengths of these *BrNRT1* protein sequences varied from 194aa (BraA05g001720.3C) to 1071aa (BraA02g020750.3C) and their molecular weights ranged from 21.5 kD (BraA05g001720.3C) to 117.9 kD (BraA02g020750.3C), while their theoretical isoelectric point (pI) values ranged from 5.23 (BraA09g059270.3C) to 9.58 (BraA06g009470.3C). Through a subcellular localization prediction, it was revealed that a majority of *BrNRT1* is located in the plasma membrane.

### 2.2. Phylogenetic Analysis of BrNRT1s

To examine the evolutionary relationships within *BrNRT1s*, a multi-sequence alignment was performed, using the 84 *BrNRT1s* to construct a phylogenetic tree (Figure 1). Then, the information gathered from the alignment was utilized to construct an unrooted phylogenetic tree. Notably, the absence of any *BrNRT1* from the phylogenetic tree construction resulted from the deletion of either the C-terminal or N-terminal region. The 84 *BrNRT1s* were divided into six distinct subfamilies (Group 1–Group 6). Interestingly, subfamily Group 2 (21) has the highest number of NRT genes among these subfamilies, while subfamily Group 1 (3) has the lowest number of NRT genes. All members of the subfamily contained whole structural domains (Figure 1).

### 2.3. Gene Structure and Conserved Motif Analyses of BrNRT1s

Determining the gene structure and conducting a conserved motif analysis was beneficial for our understanding of the NRT1 family. A gene structural analysis of *BrNRT1s* was conducted using the GSDS tool. *BrNRT1* members with close relationships shared similar gene structures. The MEME program was used to predict and analyze the motif compositions of the 84 members BrNRT1 family. As shown in Figure 2, there were 12 distinct motifs found. Of the 84 BrNRT1 proteins, 53.6% contained twelve conserved motifs, while the remaining proteins were absent in one or more motifs. Furthermore, it was found that BrNRT1 members with close relationships also shared similar motif compositions and orders. For instance, motif 10 was absent in five *BrNRT1* genes belonging to the V subfamily. Figure 2 shows that none of the seven *BrNRT1* genes in the Group 2 subfamily contained motif 9.

### 2.4. Regulatory Mechanism in the Promoter Regions of BrNRT1 Genes

Eleven cis-acting regulatory elements (CREs) from the promoters of the 84 *BrNRT1* genes were discovered using PlantCARE (http://bioinformatics.psb.ugent.be/webtools/plantcare/html/, accessed on 15 July 2023), as previously described by Hu et al. [34] (Figure 2). The study of CREs can provide valuable insights into the probable functionalities of *BrNRT1* genes. The study revealed that each *BrNRT1* gene contained one or more CREs in their promoter region. Additionally, many closely related genes exhibited comparable forms of CREs. As shown in Figure 2, the CREs discovered in the promoter region of *BrNRT1* are known to be involved in protein transport. Members of the Major Facilitator Superfamily (MFS) can be found in all BrNRT1 genes (Figure 2).

Furthermore, it is notable that BraA09g012480.3C exhibited the presence of a GAG-pre-integrase domain (gag_pre-integrs (pfam13976)), suggesting the presence of mobile genetic elements. Another relevant gene is *BraA07g034710.3C*, an SPFH superfamily (SPFH_like_u4 (cd03407)). In *BraA07g034710.3C*, a potential association of genes with lipid rafts was found. Specifically, the gene *BraA03g019780.3C* was identified in this context. The *BraA03g019780.3C* protein possessed the PTZ00169 superfamily (cl36523), an ADP/ATP transporter located on the adenylate translocase (Figure 2).

### 2.5. Expression Patterns of BrNRT1s in Different Tissues

To further characterize the role of *BrNRT1s* in Chinese cabbage, we investigated the expression levels of *BrNRT1s* in two tissues (young leaves and tender roots) and at two nitrogen levels (0.2 mM and 6.0 mM). We analyzed the expression patterns of *BrNRT1* genes in two Chinese cabbages (L14 and L40). Fifty-two (62%) of the eighty-four *BrNRT1* genes were found in the tissues of both L14 and L40 (Appendix A). We found 52 (62%) and 55 (65%) *BrNRT1s* in the tissues of both the LN and CK treatment cabbages, respectively. Meanwhile, 46 (55%) *BrNRT1s* were found in the root, whereas 54 (64%) were found in the leaves (Appendix A).

The *BrNRT1* genes exhibited distinct patterns of expression in several subfamilies. In the Group 6 subfamily, the expression levels of most *BrNRT1* genes increased significantly in various tissues (FPKM > 1). Most genes belonging to the Group 4 and Group 5 subfamilies exhibited reduced expression levels (FPKM < 1). Interestingly, several *BrNRT1* genes showed tissue-specific expression patterns. For instance, most *BrNRT1* genes were more highly expressed in the root tissue than in the leaf tissue. The different expression patterns in each tissue demonstrated various functions of *BrNRT1s* in the whole plant (Appendix A).

### 2.6. The Expression of BrNRT1s in Response to LN Stress

Nitrate is a major substrate for NRT1-protein-mediated transport. In the roots of L14 and L40, the presence of N in the nutrient solution altered the expression of specific nitrogen utilization genes. This is crucial for N uptake, translocation, and assimilation. It was found that some of *BrNRT1s*’ expression levels differed significantly under different nitrate concentrations (Appendix A).

We identified differentially expressed genes (DEGs) between two Chinese cabbages (L14 and L40) and different nitrate concentrations. In L14, 23 DEGs related to *BrNRT1* were identified in response to LN. Seven DEGs were up-regulated and three DEGs were down-regulated in LN14L (L14-LN leaves) vs. CK14L (L14-CK leaves). Moreover, 20 DEGs were found in LN14R (L14-LN root) vs. CK14R (L14-CK root) of which 14 DEGs were up-regulated and the other six DEGs were down-regulated. In LN40L (L40-LN leaves) vs. CK40L (L40-CK leaves), 37 DEGs related to *BrNRT1* were identified in response to LN levels. Among these DEGs, nine DEGs were up-regulated, while eleven DEGs were down-regulated. In addition, there were 13 up-regulated and 13 down-regulated DEGs in LN40R (L40-LN root) vs. CK40R (L40-CK root) (Appendix A and Figure 3).

### 2.7. The Involvement of BrNRT1 DEGs in Response to LN Stress

The presence of N in the absorption, translocation, and assimilation of N crucially influences the expression of some *BrNRT1* genes in L14 and L40. As shown in Table 1, some of these gene expression levels differed significantly between the shoot and the root.

Several genes, such as NRT1.1 (BraA08g031180.3C), NPF2.10 (BraA01g024830.3C), NPF4.1 (BraA03g041550.3C), NPF4.3 (BraA09g017570.3C), NPF6.4 (BraA01g032090.3C), and NPF8.1 (BraA09g045140.3C), were expressed in the root and the shoot; however, the expression levels differed significantly in L40. Interestingly, NRT1.1 (BraA09g060970.3C), NRT1.11 (BraA06g002150.3C), NPF2.10 (BraA06g019030.3C), and NPF8.3 (BraA09g059270.3C) differentially only in the root of L40. Contrastingly, NRT1.7 (BraA02g019460.3C), NRT1.9 (BraA09g056980.3C), NPF4.1 (BraA03g041540.3C), and NPF4.4 (BraA09g031580.3C) were highly expressed only in the leaves of L40. Similarly, NRT1.5 (BraA05g023200.3C) and NRT1.8 (BraA03g049670.3C) were highly expressed specifically in the leaf tissues of the L14 cabbage. Furthermore, NPF3.1 (BraA02g018550.3C) was highly expressed in the root tissues of L14 and L40 (Table 1).

*NRT1.8* (*BraA01g012370.3C*), *NRT1.1* (*BraA06g009470.3C*), and *NPF4.5* (*BraA07g013380.3C*) were highly expressed in both cultivars under LN stress. Notably, L14 and L40 had comparable regulatory pathways for these genes. Furthermore, the expression of *NPF2.11* (*BraA09g007360.3C*) showed a similar pattern in both cultivars except in the shoot of the L14 cultivar. The *NPF6.4* (*BraA01g025490.3C*) gene was only detected in the roots of L14 plants treated with a LN concentration. In addition, *NRT1.9* (*BraA09g056980.3C*) was only detected in the shoots of L40 plants under LN stress. These findings demonstrate that *BrNRT1s* have different expression patterns in specific tissues and cultivars (Table 1).

### 2.8. The Expression Profiles of BrNRT1s in the N Metabolism Pathway under LN Stress Conditions

Nitrate is the primary substrate for NRT1 protein transport. In this study, we examined the expression profiles of 24 *BrNRT1* genes in response to varying nitrate concentrations. A total of 24 *BrNRT1* genes were selected from the six subfamilies to elucidate the expression patterns of the genes across various subfamilies (Figure 4).

The results of a multiple-factor analysis indicate that the presence of *BrNRT1* were significantly different under different nitrogen treatments: genotypes, tissue, genotype*treatment, genotype*tissue, treatment*tissue, and genotype*treatment*tissue (*p* < 0.01) (Appendix A). As we presented in Appendix A, the relative average values of all *BrNRT1* genes are significantly different under two nitrogen treatments, indicating that low nitrogen affected the reduction of *BrNRT1s*. Interestingly, the average values of *BraA08g031180.3C*, *BraA06g009470.3C*, and *BraA09g060970.3C* are the highest among the *BrNRT1s* (Appendix A), implying that the low-nitrogen treatment had the greatest impact on these three genes. A total of 20 BrNRT1 genes in tissue were significantly affected by nitrogen supply levels (*p* < 0.05) under two nitrogen treatments, while 16 BrNRT1 genes were significantly affected by nitrogen supply levels (*p* < 0.01) compared to other treatments (Appendix A). A total of nine, ten, and seven genes were significantly affected by nitrogen supply levels in genotype*treatment, genotype*tissues, and treatment*tissues, respectively (*p* < 0.01). Furthermore, a total of eight genes were significantly affected by nitrogen supply levels (*p* < 0.01) in genotype*treatment*tissues.

The expression profiles of 24 representative *BrNRT1* genes under LN stress and a control are shown in Figure 4. The results showed that the levels of expression of 18 *BrNRT1*s’ transcripts were highly induced after LN stress. Fourteen *BrNRT1s* were up-regulated by twofold or more in the shoot of L14 and an additional fourteen *BrNRT1s* were up-regulated in the root of L14. In addition, a total of ten *BrNRT1s* were up-regulated in the shoot of L40, while four *BrNRT1s* were up-regulated by twofold or more in the shoot of L40. Significant differences were found under LN stress conditions in which one *BrNRT1* was detected in the shoot of L14, nine *BrNRT1s* were detected in the root of L14, eight *BrNRT1s* were detected in the shoot of L40, and four *BrNRT1s* were detected in the shoot of L40 (Figure 4 and Appendix A).

In addition, many *BrNRT1s* showed tissue- and variety-specificity expression patterns, consistent with the RNA-seq analysis results. The results revealed that a total of seven *BrNRT1s* were predominantly expressed in shoots, whereas ten *BrNRT1s* were primarily expressed in roots. Several genes, such as *BraA06g002150.3C*, *BraA04g005820.3C*, and *BraA09g045140.3C*, might be considered. *BraA06g002150.3C* was highly expressed in the shoot, implying its potential role in physiological activities such as photosynthesis and protein synthesis. Meanwhile, *BraA04g005820.3C* and *BraA09g045140.3C* were highly expressed in the root, suggesting that *BraA04g005820.3C* and *BraA09g045140.3C* might be involved in the long-distance transportation of nitrate from the root to any location in the whole plant (Figure 4 and Appendix A).

The number of DEGs in the shoot increased dramatically in the L40 (eight *BrNRT1*s) compared with L14 (one *BrNRT1*). In contrast, the number of DEGs in the root of L40 (four *BrNRT1s*) was lower than the number of DEGs in L14 (seven *BrNRT1s*). Notably, the *BraA06g009470.3C* genes were the most highly expressed in the L40 shoot. These findings illustrated that the *BraA06g009470.3C* gene transcript levels significantly increased in the shoot of L40 under LN stress conditions. However, *BraA06g009470.3C*’s response to LN stress in L14 was slightly low. Contrastingly, the level of *BraA10g026590.3C’s* transcript expression was elevated in response to LN stress in the L14 root but displayed a slightly elevated response in the L40 root (Figure 4 and Appendix A).

## 3. Discussion

The *NRT1* gene is widely distributed in eukaryotes and has been confirmed to be involved in the transport of nitrate/nitrite [14], dipeptide/tripeptide [35], and various hormones [36,37,38]. In recent years, extensive research has been conducted on the *NRT1* gene in Arabidopsis, rice, and other plant species. However, studies about the *NRT1* gene in Chinese cabbage are still limited. However, the production of Chinese cabbage is important as it is one of China’s primary vegetables. This cultivation plays a crucial role in providing a stability domestic vegetable supply and promoting sustainable agricultural development across the country [31]. This study identified a total of 84 *BrNRT1* genes within the genome of Chinese cabbage. These genes were subsequently categorized into six subgroups based on the similarity of their structures. Our findings showed that Chinese cabbage has a different number of *NRT1* subgroups compared to other species, including *Spirodela polyrhiza*, *Poncirus trifoliata*, and *Setaria* sp. under abiotic stress, illustrating the distinct response of *NRT1* members in coping with stress cues [39,40].

The changes in gene structure might be caused by gene mutation and structural variants. Structural variants might influence the evolution of gene families. Based on this investigation of the *BrNRT1* gene’s structure and conserved motifs results, the gene’s structure is highly conserved. In the same subfamily, closely related genes show similar gene structures and motif deletions. Similar studies were also reported in *Arabidopsis*, *Populus trichocarpa*, soybean [41], and *Brassica napus* [42]. We assumed that the diversity of gene functions in the genes of the Chinese cabbage family might be related to gene structure alterations and conserved motifs.

The structure and function of *BrNRT1* genes were consistent; in particular, the expression of closely related *BrNRT1* genes in the same subfamily showed similar patterns. However, the expression of some *BrNRT1* genes showed specificity for different tissues. Numerous genes are frequently expressed in certain tissues and have tissue-specific functions [43]. For example, *NRT1.7* (*BraA02g019460.3C*), *NRT1.9* (*BraA09g056980.3C*), *NPF4.1* (*BraA03g041540.3C*), and *NPF4.4* (*BraA09g031580.3C*) were predominantly expressed in the leaf. *NRT1.1* (*BraA09g060970.3C*), *NRT1.11* (*BraA06g002150.3C*), *NPF2.10* (*BraA06g019030.3C*), and *NPF8.3* (*BraA09g059270.3C*) were predominantly expressed in the root. The expression patterns of *NRT1* genes in Chinese cabbage varied between L14 and L40. For instance, *NRT1.1* (*BraA08g031180.3C*), *NPF2.10* (*BraA01g024830.3C*), *NPF4.1* (*BraA03g041550.3C*), *NPF4.3* (*BraA09g017570.3C*), *NPF6.4* (*BraA01g032090.3C*), and *NPF8.1* (*BraA09g045140.3C*) were only highly expressed in L40. Meanwhile, *NRT1.5* (*BraA05g023200.3C*), *NRT1.8* (*BraA03g049670.3C*), *NPF3.1*(*BraA07g030700.3C*), and *NPF6.4* (*BraA05g025490.3C*) were highly expressed in L14. Our findings are in accordance with previous studies in *B. rapa*, *Poncirus trifoliata*, and *Spirodela polyrhiza* [39,44,45]. Hence, we assumed these genes could provide valuable allelic variations for the enhancement of nutrient use efficiency (NUE) in Chinese cabbage.

The functions of NRT1.1 are complex and include acting as a nitrate protein transporter, root development regulator, auxin activity transporter, nitrate signal transmitter, nutrient signal inducer, and plant microecological interaction regulator [22]. In this study, we found that three *BrNRT1.1* (*BraA06g009470.3C, BraA08g031180.3C, and BraA09g060970.3C*) transcript levels were up-regulated under LN conditions, illustrating that *BrNRT1.1* has similar effect on nitrate transport. Similarly, a previous study also stated that *NRT1.1* acts as a nitrate transport protein in numerous plant species under abiotic stresses [1].

Generally, plants require the ability to acclimate to challenging conditions, synchronize their growth processes, and regulate functionally related genes. The *NRT1.8* genes (*BraA01g012370.3C* and *BraA03g049670.3C*), homologs of *AtNPF7.2*/*AtNRT1.8,* were significantly up-regulated under LN conditions in L14. Similarly, the *NRT1.5* gene (*BraA05g023200.3C*), a homolog of *AtNPF7.3*/*AtNRT1.5*, was down-regulated considerably under LN conditions in L14. Contrastingly, two members of the NPF7 family (*AtNPF7.3*/*AtNRT1.5* and *AtNPF7.2*/*AtNRT1.8*) showed opposite functions under cadmium stress compared to other stress conditions, including LN stress [46]. These findings illustrate that *NRT1s*, which function in nitrate transporters, have different functions in response to specific stresses. Our findings are in accordance with previous studies which stated that NRT1 is important in the absorption and transportation of N for coping with various stresses in plants [47,48].

According to previous reports, five NPF proteins, namely NPF2.10 (GTR1), NPF2.11 (GTR2), NPF2.9 (NRT1.9), NPF2.14, and NPF2.13 (NRT1.7), have dual functions in terms of facilitating the transport of nitrate–nitrogen at low-affinity levels and the transport of erucic acid. The multiple functions of NPF proteins play crucial roles in improving nutrition uptake and plant defense mechanisms [24,34,49]. NPF homologous genes in Chinese cabbage were preferentially expressed in L40.

In addition, NRT1 also has the capability to transport hormone substrates. Our findings showed that NPF proteins (AtNPF4.1 (AIT3) and AtNPF4.5 (AIT2)) are involved in ABA and GA transport activities [50]. In Arabidopsis, a total of nine and eighteen *NRT1* members actively regulate ABA and GA transport, respectively [36]. Numerous studies have shown that the expression patterns of these genes are usually closely related to their CREs [47]. This study discovered various cis-elements involved in protein transport in the *BrNRT1* promoter, such as the MFS family (84/84 genes), which is involved in extensive substrate movement in biofilms. This finding indicates that *BrNRT1* plays a vital role as a protein transporter in Chinese cabbage [48,51]. The transcript levels of *NPF4.1* (BraA03g041540.3C) and *NPF4.5* (BraA07g013380.3C) were significantly down-regulated under LN conditions. Moreover, previous study revealed that nitrate interacts with phytohormones like auxins, cytokinins, ABA, GA, and ethylene. According to [39], it was shown that hormone production, de-conjugation, transport, and signaling are partially regulated by nitrate signaling.

Considering the fact that N is an important nutrient for plants, NPFs (NRT1/PTR Family6) also facilitate the transportation of dipeptides, amino acids, and nitrate, which are the primary forms of nitrogen utilized by plants for their nutritional needs [52]. The dipeptide transport activity of two previously identified dipeptide transporters (AtNPF8.1 and AtNPF8.3) was confirmed in Arabidopsis [53]. The homologous genes, namely *NPF8.1* (*BraA09g045140.3C*) and *NPF8.3* (*BraA09g059270.3C*)*,* were significantly down-regulated under LN conditions and preferentially expressed in L40, demonstrating that these genes contribute to providing nutrition in Chinese cabbage. In addition, our findings demonstrated that treatments, genotypes, and tissues are significantly affected the DEGs in Chinese cabbage (*p* < 0 01). The highest number of DEGs was identified in tissue, indicating NRT1 is susceptible to different nitrogen treatments in different tissues of Chinese cabbage. Also, several NRT1 genes are subject to dual or triple regulation simultaneously. We assumed that the *BrNRT1* gene’s mechanism is complex in response to LN conditions in Chinese cabbage. In this study, the expression of several *BrNRT1* genes changed significantly, demonstrating the involvement of these genes in the uptake of nitrate and other substances.

## 4. Conclusions

In this study, a total of 84 *BrNRT1* genes were identified at the genomic level, and their gene structures, conserved motifs, cis-acting elements, and expression patterns were analyzed. The gene structure and motif of the *BrNRT1* protein in each subfamily are highly conserved, indicating that these genes might be involved in regulating similar functions in response to LN stress. Furthermore, our findings show that the expression pattern of the *NRT1* genes in Chinese cabbage is tissue-specific and variety-specific. An RT-qPCR analysis showed that most selected *BrNRT1* genes were induced by nitrate and involved in N uptake and transport. These findings provide basic information for future research and a promising strategy for Chinese cabbage breeding programs to improve N uptake efficiency and yield productivity.

## 5. Materials and Methods

1.Plant Material and Treatment

Two inbred lines of Chinese cabbage named L14 (LN-sensitive) and L40 (LN-tolerant) were analyzed in the present study. These two cultivars were screened out according to their agronomic trait yields under a normal N supply (N: 270 kg/ha) and LN (N: 54 kg/ha) conditions. The seedlings were obtained from the Shandong Academy of Agricultural Sciences, China, and selected for this study. The seedlings were cultured in a full nitrogen hydroponic solution for 2 weeks. Then, we transferred the seedlings to a low-nitrogen (0.2 mM, LN ) nutrient solution for 1 week as a N-starvation treatment. Full nitrogen was used as the control (6 mM, CK) [54]. The shoots and roots were harvested, rapidly frozen in liquid nitrogen, and then stored at −80 °C. The specific formulation of the medium is shown in Table 2.

2.RNA Isolation:

The total RNA was isolated using Trizol (Invitrogen, Carlsbad, CA, USA). We used a 1% agarose gel and a 2100 Bioanalyzer RNA Nanochip (Agilent, Santa Clara, CA, USA) to measure the quality of the total RNA. Then, we used a NanoDrop ND-2000 Spectrophotometer (Nano-Drop, Wilmington, DE, USA) to measure the concentration of the total RNA.

3.cDNA Library Construction, Sequencing, and Data Processing:

Equal quantities of total RNA from three replicates were mixed according to the manufacturer’s protocol (Illumina Inc., San Diego, CA, USA). DNA library construction, sequencing, and data processing and the construction of cDNA libraries occurred at Novogene (Tianjin, China). First, the Agilent 2100 Bioanalyzer (Agilent Technologies Inc., Santa Clara, CA, USA) and ABI StepOnePlus Real-Time PCR System (Applied Biosystems, Inc., Foster City, CA, USA) were used to qualify and quantify the sample library. Then, an IlluminaHiSeq™ 2000 (Illumina Inc., San Diego, CA, USA) was used to perform high-throughput sequencing. Finally, the original image data were generated by the sequencer and were transferred into raw sequence data via base calling. High-quality, clean data were obtained from the raw sequence data after removing empty reads, adapters, and low-quality reads. The clean reads were mapped to the *Brassica rapa* L. ssp. *pekinensis* reference genome (http://brassicadb.cn, accessed on 15 July 2023), using SOAPaligner/soap2 [55], with ≤2 mismatches being allowed in the sequence alignment to obtain unambiguous, clean tags.

4.The Identification of Chinese cabbage *NRT1* Genes:

We used the TARI website (http://www.arabidopsis.org/, accessed on 15 July 2023) to obtain 77 NRT1 sequences of *A. thaliana*. The Brassica database (BRAD) website (http://www.brassicadb.cn/#/, accessed on 15 July 2023) was used to download Chinese cabbage genome sequences. The putative *NRT1s* of Chinese cabbage were obtained via screening Chinese cabbage genome sequences via a BlastP search, using *AtNRTs* as queries. In this study, we encountered statistical models known as hidden models. We identified members of the gene family using the online sequence analysis tool named HMMER (http://hmmer.org/download.html, accessed on 15 July 2023). HMMER implementation uses probabilistic models known as profile-hidden Markov models (profile HMMs) to search sequences of homologs in the databases and then align the sequences [56]. The putative members of the *NRT1* gene family obtained from the two approaches were merged to generate a unique gene list. To confirm the presence of the NRT1 domain, we used the protein sequences of putative *NRT1* gene family members of Chinese cabbage in the Pfam (http://pfam.xfam.org/, accessed on 15 July 2023) and SMART (http://smart.embl-heidelberg.de/, accessed on 15 July 2023) databases for analysis.

5.The Physicochemical Properties of the Proteins:

The molecular masses of the putative NRT1 proteins were calculated using the Compute pI/Mw tool of ExPaSy (http://web.expasy.org/compute_pi/, accessed on 15 July 2023). The number of amino acids, the molecular weight (MW), and the theoretical isoelectric point (pI) were computed using the ProtParam tool (https://web.expasy.org/protparam/, accessed on 15 July 2023) based on the general feature format (GFF) file of the *NRT1* genes in *Brassica rapa* L. ssp. *pekinensis*.

6.The Construction of a Phylogenetic Tree:

Annotation of the Chinese cabbage genome was downloaded from the Brassica database (BRAD, http://www.brassicadb.cn/#/, accessed on 15 July 2023), including the complete Brassica A genome sequence from *B. rapa* (Chiifu-401). The *NRT1* nucleotide sequences were aligned using SnapGene 5.0.5 Software (from GSL Biotech; available at snapgene.com). Schematic gene structure diagrams were drawn using the exon/intron organization of the *NRT1* genes and determined using the Gene Structure Display Server (http://gsds.cbi.pku.edu.cn/, accessed on 15 July 2023) and a GFF file, as described by Guo et al. [57]. The motifs of the NRT1 protein sequences were analyzed using MEME online tools (http://meme-suite.org/tools/meme, accessed on 15 July 2023). For the phylogenetic analysis, protein sequences of *NRT1* from Chinese cabbage, Arabidopsis, and rice were aligned with MUSCLE multiple sequence alignment tools from SnapGene5.0.5 Software (from GSL Biotech; available at snapgene.com), using thedefault settings. The phylogenetic tree was constructed using the Neighbour-Joining (NJ) method and a bootstrap test was conducted using MEGA 7.0 Software with the following parameter settings: pairwise deletion mode, Poisson correction, and bootstrapping (1000 replicates).

7.Gene Structure and Conserved Motif Analyses:

We used the BRAD database (http://brassicadb.org/brad/, accessed on 15 July 2023) to analyze the gene structure of Chinese cabbage. TBtools software was used to display the distribution patterns of the exons and introns of *NRT1* gene and motif results [58]. An online prediction of the conserved motif of the *NRT1* gene encoding protein was conducted with the maximum number of motif discoveries set to 20 and the other parameters set to default values. 

8.Analysis of cis-acting Elements:

First, we extracted a 2000 bp promoter region sequence upstream of each *BrNRT1 member’s* starting codon ATG using TBtools software. Then, the PlantCARE database (http://bioinformatics.psb.ugent.be/webtools/plantcare/html/, accessed on 15 July 2023) was used to perform a cis-acting element analysis. Finally, we used TBtools software to visualize a distribution map of cis-acting elements in the *BrNRT1* members’ promoter regions.

9.Analysis of *BrNRT1* Genes’ Expression Profiles in Response to LN Stress:

The gene expression levels of all of *BrNRT1* genes were normalized according to the RPKM (reads per kb per million reads) method [59]. Differentially expressed *BrNRT1* genes (DEGs) between the LN-treated groups (L14–LN and L40–LN) and control groups (L14–CK and L40–CK) were screened based on an algorithm developed by Audic and Claverie [60]. We used the threshold “FDR ≤ 0.001 and the absolute value of log_2_ Ratio ≥ 1” to judge the significance of the *BrNRT1* gene expression difference. A Kyoto Encyclopedia of Genes and Genomes (KEGG) pathway analysis was performed to investigate the high-level functions and utilities of the biological system. The cluster Profiler R package was used to test the statistical enrichment of DEGs in KEGG pathways.

10.Reverse Transcription and RT-qPCR Detection:

A TOYOBO ReverTra Ace qPCR RT Kit (FSQ-101, TOYOBO, Osaka, Japan) was used to synthesize the first-strand cDNA. We performed a real-time quantitative PCR (RT-qPCR) analysis using a 2 × M5 HiPer SYBR Premix EsTaq plus (with Tli RNaseH) (MF787, Mei5bio, Beijing, China) and an IQ5 Real-Time PCR System (BIO-RAD, Hercules, CA, USA). The PCR reactions were performed at 95 °C for 30 s, followed by 40 reaction cycles (95 °C for 5 s, followed by 60 °C for 20 s). For each gene, three independent PCR reactions were carried out. Then, the relative gene expression was calculated using the 2^−∆∆CT^ method [61]. The gene-specific primers were designed using Primer Premier 5.0, as shown in Table 3.

## 6. Data Processing and Analysis

Microsoft Excel (Version 2016, Microsoft Corporation, Washington, DC, USA) and SPSS (Version 13.0., SPSS Inc., Chicago, FL, USA) statistical software were used for the statistical analysis and to generate the diagrams, as described by Anwar et al., 2020 [62]. A univariate analysis was used to test for significant differences in the relative gene expression levels of the BrNRT1s according to the results of a qPCR among the genotypes, tissues, treatments, and their interactions. All data are expressed as means ± standard deviations (SDs).

## Figures and Tables

**Figure 1 plants-12-03882-f001:**
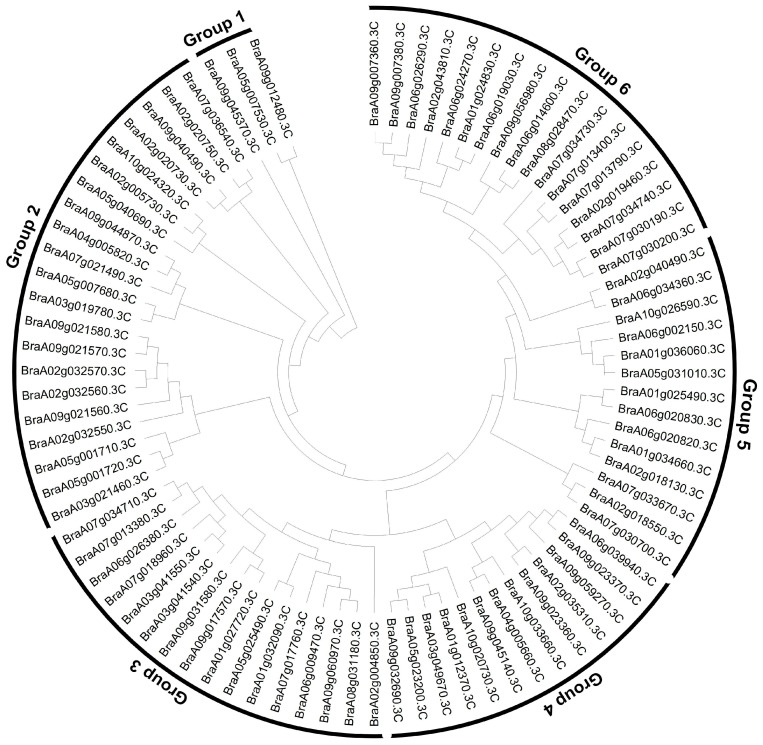
The unrooted phylogenetic tree of NRT proteins in Chinese cabbage.

**Figure 2 plants-12-03882-f002:**
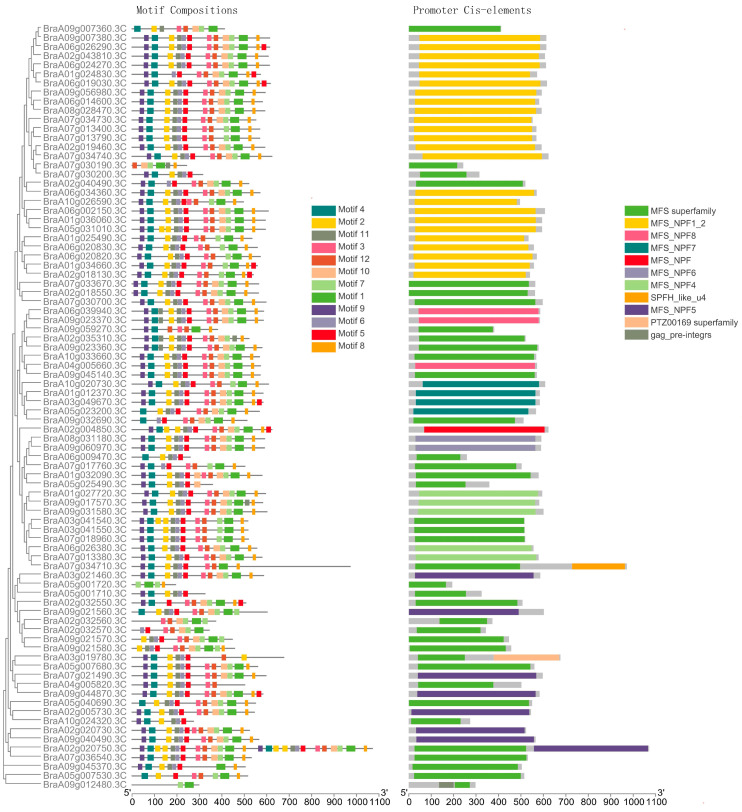
Gene structure and conserved protein motif organization of *NRT1* genes in Chinese cabbage.

**Figure 3 plants-12-03882-f003:**
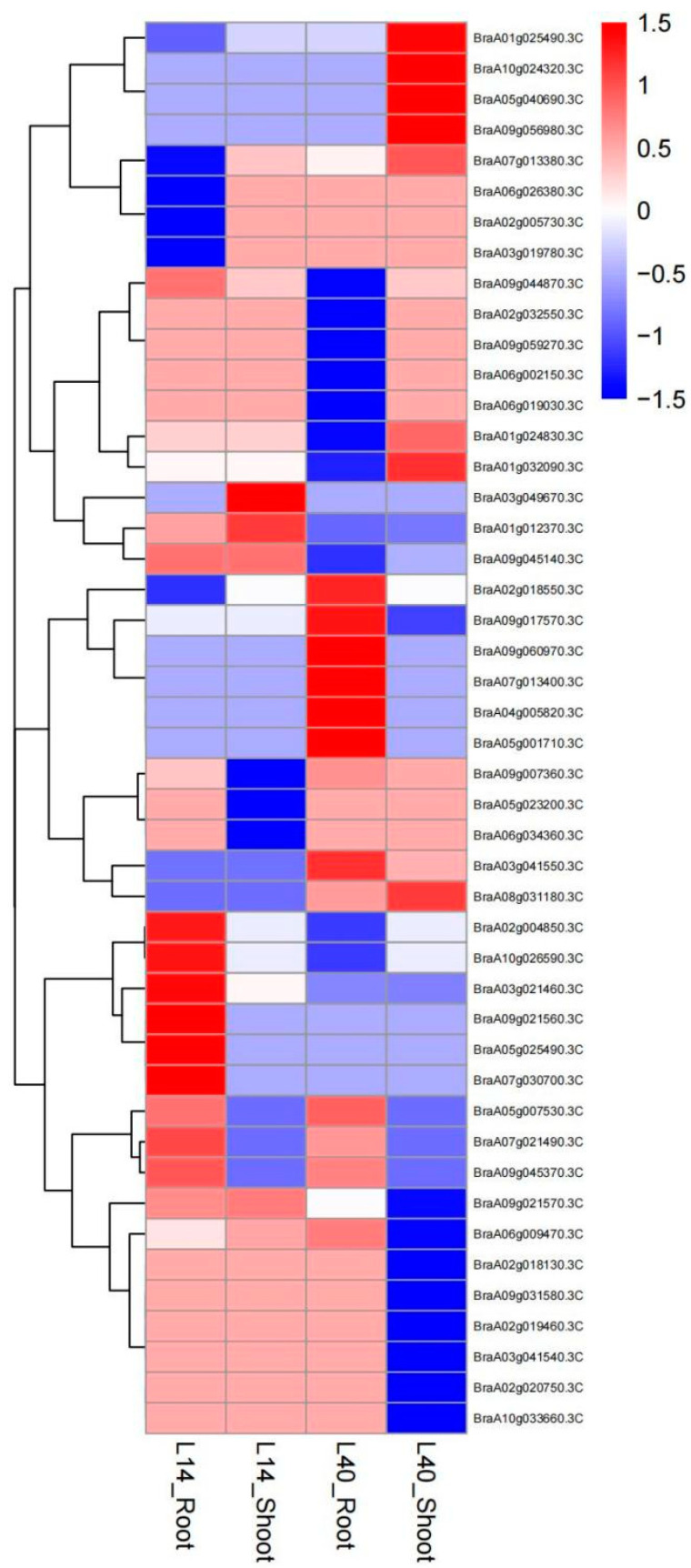
Expression heatmap of *BrNRT1* genes in Chinese cabbage.

**Figure 4 plants-12-03882-f004:**
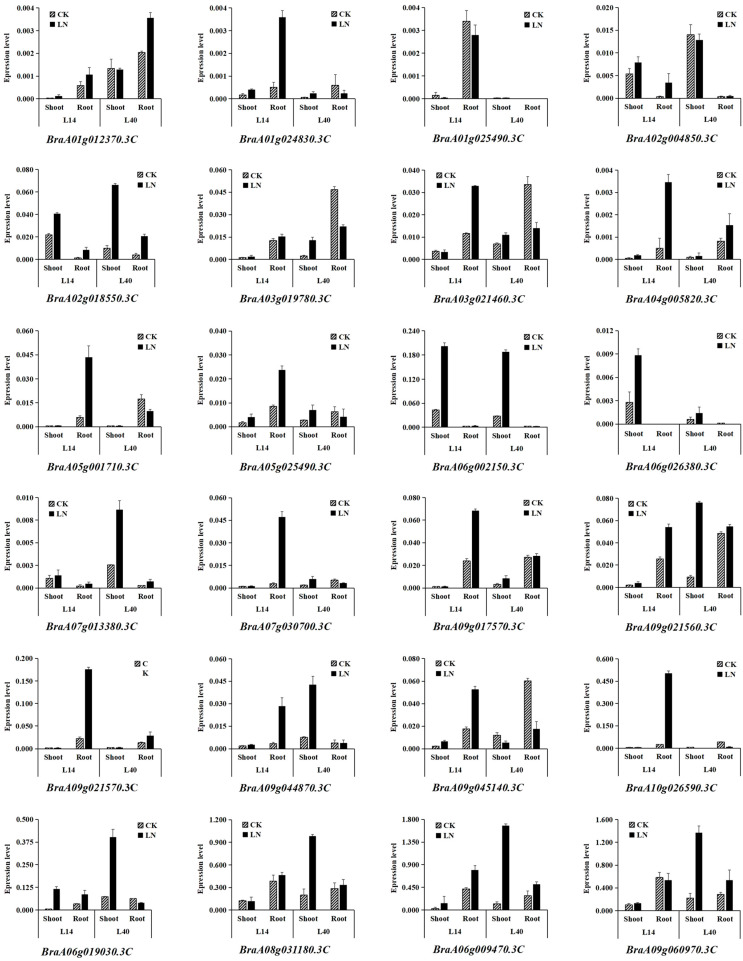
qRT-PCR validation of DEGs from high-throughput sequencing analyses. Relative expression levels were calculated as a reference gene using the formula 2^−∆∆CT^. The values indicated are the means of three biological replicates ± SEs.

**Table 1 plants-12-03882-t001:** DEGs identified related to LN stress conditions.

	Gene ID	Chr.	Start	End	log_2_ (LN/CK)			
					Shoot		Root	
					L14	L40	L14	L40
NRT1.1	*BraA06g009470.3C*	A06	5,167,089	5,168,324	1.73	ns	1.42	1.94
NRT1.1	*BraA08g031180.3C*	A08	20,956,829	20,960,518	ns	1.55	ns	1.13
NRT1.1	*BraA09g060970.3C*	A09	42,614,336	42,618,089	ns	ns	ns	1.31
NRT1.5	*BraA05g023200.3C*	A05	17,088,575	17,091,980	−2.86	ns	ns	ns
NRT1.7	*BraA02g019460.3C*	A02	11,150,404	11,157,116	ns	−2.42	ns	ns
NRT1.8	*BraA01g012370.3C*	A01	6,414,144	6,416,171	2.29	−1.28	1.19	−1.5
NRT1.8	*BraA03g049670.3C*	A03	25,372,935	25,374,941	2.61	ns	ns	ns
NRT1.9	*BraA09g056980.3C*	A09	40,666,918	40,669,296	ns	1.8	ns	ns
NRT1.11	*BraA06g002150.3C*	A06	1,283,012	1,287,130	ns	ns	ns	−1.81
NPF2.10	*BraA01g024830.3C*	A01	14,570,611	14,572,963	ns	1.27	ns	−3.49
NPF2.10	*BraA06g019030.3C*	A06	10,801,268	10,803,925	ns	ns	ns	−1.1
NPF2.11	*BraA09g007360.3C*	A09	4,260,320	4,262,091	ns	1.91	1.75	2.04
NPF3.1	*BraA02g018550.3C*	A02	10,436,201	10,443,396	ns	ns	−1.56	1.7
NPF3.1	*BraA07g030700.3C*	A07	22,416,610	22,420,494	ns	ns	1.69	ns
NPF4.1	*BraA03g041540.3C*	A03	20,827,737	20,829,854	ns	−2.4	ns	ns
NPF4.1	*BraA03g041550.3C*	A03	20,839,549	20,842,211	ns	1.8	ns	2.81
NPF4.3	*BraA09g017570.3C*	A09	10,925,112	10,930,083	ns	−1.02	ns	1.46
NPF4.4	*BraA09g031580.3C*	A09	24,492,209	24,495,098	ns	−3.07	ns	ns
NPF4.5	*BraA07g013380.3C*	A07	12,534,292	12,537,206	−1.01	ns	−3.75	−1.43
NPF6.4	*BraA01g025490.3C*	A01	15,182,959	15,184,900	ns	4.67	−1.92	ns
NPF6.4	*BraA01g032090.3C*	A01	21,885,380	21,889,127	ns	1.31	ns	−1.49
NPF6.4	*BraA05g025490.3C*	A05	19,200,397	19,202,444	ns	ns	1.28	ns
NPF8.1	*BraA09g045140.3C*	A09	34,557,783	34,560,452	ns	−1.21	ns	−1.86
NPF8.3	*BraA09g059270.3C*	A09	41,823,598	41,826,639	ns	ns	ns	−1.85

Note: “ns” represents that there was no significant difference in the gene expression levels between the shoot and the root.

**Table 2 plants-12-03882-t002:** The hydroponic nutrient solution content for each treatment.

Nutrient Solution Formula		Treatment	
		CK (6.0 mM)	Low N (0.2 mM)
ME	KNO_3_	3 mM	0.1 mM
	KCl	0.0 mM	3 mM
	KH_2_PO_4_	1.5 mM	1.5 mM
	K_2_HPO_4_	1.5 mM	1.5 mM
	(NH_4_)SO_4_	0.75 mM	25 mM
	MgCl_2_	1 mM	1 mM
	K_2_SO_4_	2 mM	2 mM
	Ca(NO_3_)_2_	0.75 mM	0.75 mM
	CaCl_2_	3.25 mM	3.25 mM
Fe	Na_2_Fe-EDTA	40 µM	40 µM
TE	H_3_BO_3_	60 µM	60 µM
	MnSO_4_	14 µM	14 µM
	ZnSO_4_	1 µM	1 µM
	CuSO_4_	0.6 µM	0.6 µM
	NiCl_2_	0.4 µM	0.4 µM
	H_2_MoO_4_	0.3 µM	0.3 µM
	CoCl_2_	20 nM	20 nM

Note: ME: macroelement; Fe: ferrum; TE: trace element.

**Table 3 plants-12-03882-t003:** Sequences of primers used in the study.

Genes	Forward Primer	Reverse Primer
Bra-Action	ATACCAGGCTTGAGCATACCG	GCCAAAGAGGCCATCAGACAA
*BraA01g012370.3C*	GAAGGTGGAGAGTGGATCAAC	AAGAGTGAAGCCATCTGAGTG
*BraA01g024830.3C*	TGGCATCGCTCGTGTTATAG	CTGGTCCGAGTATTTGAGTGTAG
*BraA01g025490.3C*	GTCATAGTCGGAATTGGAGAGG	TGTAGAAGCATATCCCAATCACC
*BraA02g004850.3C*	TCTGTCTTCGTCCCTATCACC	GCGTAATGTCTCCTGTAGTTCTC
*BraA02g018550.3C*	AGATATGTGAGAAACTGGCGG	AGTTAGTGAGAGTGTTGGCTG
*BraA03g019780.3C*	TTGGTTGTGGAAGTCTAGCTG	TGAATGTCCCAAAGAGTCCTG
*BraA03g021460.3C*	AGTCACCTGAGGAAATGCAG	TTATCCCCAATCCAACTCTTCC
*BraA04g005820.3C*	CTGAAATCTATGGAGTCGCCG	TGGCAACATCCGTGAGAATAG
*BraA05g001710.3C*	ACGTTTAGCTCCTTTTCGGG	TGCC ATAACCTAAACCCCAC
*BraA05g025490.3C*	TCTGGGAGTGTTGCTGTT	GATGGTGAAGTCTCCTGAAGTC
*BraA06g002150.3C*	TGTACTTATCGCGTTCACTGG	CTTGCTCACATTGCGTCTAAC
*BraA06g026380.3C*	TCGGTGTTGGAGGTATAAAAGG	CGACCGTGACCGCTATTAAG
*BraA07g013380.3C*	TTCTCACAATCCAAGCCCG	ACAGTCCCACAAATAGAAGCG
*BraA07g030700.3C*	AACACTCTCACTAACTTCGCC	TCGTAAGCAATGTCATCCCG
*BraA09g017570.3C*	TCATCTTCCTCGTCCCTCTC	CAGTCCAAATCCTATCCGAGTC
*BraA09g021560.3C*	TGGCACTGAGAGGTATAGACTC	AAGAACGAATCAGCCATACCC
*BraA09g021570.3C*	AACTTTAGCATCCCGCtAG	CTCCCATTCGTTGAAGCAAAG
*BraA09g044870.3C*	GTCATGGCTACTTCGCTTTTG	ATCATTGGGACGAGGAAGATG
*BraA09g045140.3C*	ACCAGTTCATTGTCCCCTTC	CGAGACCTATTCCCATACGTTG
*BraA10g026590.3C*	TCCCAAGTCCATGTCAAGC	GTGTCCCTCGTTTATATCCTCTG
*BraA06g019030.3C*	CTGTCATCGCTTGTTTTCTGG	CGCTAAACCCAACATCAGAAAC

## Data Availability

Data are contained within the article and Appendix A.

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
