# Peer review of "Genome-Wide Identification of the NRT1 Family Members and Their Expression under Low-Nitrate Conditions in Chinese Cabbage (Brassica rapa L. ssp. pekinensis)"

_plants, 2023, doi:10.3390/plants12223882_

Round 1

Reviewer 1 Report

Comments and Suggestions for Authors

 Manuscript "Genome-wide identification, classification and expression analysis of the NRT1 family in Chinese cabbage (Brassica rapa L. ssp. pekinensis)" is very interesting.

Authors identified and analyzed 84 BrNRT1 genes within the Chinese cabbage genome. This study included various aspects: the phylogenic distribution, chromosomal position, gene structure, conserved motifs, and duplication pattern.
Authors conducted an analysis of the expression profile of the NRT1 gene in various tissues of Chinese cabbage exposed to varying nitrate concentrations.

Lack of "Type of the Paper".
Table 4: Number of differentially expressed genes is negative!
Figure 4: The results in Figure 4 demonstrate the analysis of a two-factor experiment. This is confirmed by the description in the methodology. Why didn't the authors present the results of the interaction between factors? This is a very important aspect in experimentation. The manuscript should be supplemented by providing a table with the results of the full analysis of variance.

In what sense was Hidden Markov Model applied? It should be detailed.
The description of "6th Data Processing and Analysis" is very weak. It should be supplemented with all the statistical methods used.

My suggestions:
"A. thaliana" - italic
"Brassica" - italic

Table S5: The Table with the results of the analysis of variance is not complete. The error information is missing. It should be corrected and completed.

Paper needs major revision.

Author Response

Comment 1: Lack of "Type of the Paper".

Response 1: Thanks for your suggestions. We have added Type of the Paper “Article” in the manuscript.

Comment 2: Table 4: Number of differentially expressed genes is negative!

Response 2: Thanks a lot for your careful reading of our manuscript. Table 4 was revised as Table 1.

Comment 3: Figure 4: The results in Figure 4 demonstrate the analysis of a two-factor experiment. This is confirmed by the description in the methodology. Why didn't the authors present the results of the interaction between factors? This is a very important aspect in experimentation. The manuscript should be supplemented by providing a table with the results of the full analysis of variance.

Response 3: Thanks for your advice. Actually, the experiment contents three factors: variety, tissue and treatment. It would be complex and messy to present all the results of the interaction between factors in the figure. For visualization and simplicity, we conducted one way ANOVA analysis using Duncan multiple comparison analysis method among all samples to show the differences between samples. In addition, we have added a new supplemental table “Table S2” to present the results of the interaction between factors.

Comment 4: In what sense was Hidden Markov Model applied? It should be detailed.

Response 4: Thanks for your suggestions. We have added descriptions on page **, line **.

Comment 5: The description of "6th Data Processing and Analysis" is very weak. It should be supplemented with all the statistical methods used.

Response 5: Thank you very much for the careful reading of our manuscript and valuable suggestions. We have rewritten the part of "Data Processing and Analysis cabbage under different N supply levels".

Comment 6: My suggestions:"A. thaliana" - italic“,"Brassica" - italic.

Response 6: Thanks for your suggestions. We have corrected the format of plant names throughout the manuscript in accordance with the reviewers' comments.

Comment 7: Table S5: The Table with the results of the analysis of variance is not complete. The error information is missing. It should be corrected and completed.

Response 7: Thanks a lot for your careful reading of our manuscript. We have corrected and completed Table S5, and it has been revised as Table S1.

Reviewer 2 Report

Comments and Suggestions for Authors

Nitrogen is a crucial element for plant growth and development. This study focuses on the identification and analysis of the NRT1 gene family to reveal the molecular mechanism of nitrate efficient absorption and utilization in Chinese cabbage. Although the significance of this study is high, some modifications efforts are needed to improve the article.

Point-to-point comments:

1. Why choose the concentrations of 0.2 mM and 6 mM for nitrogen treatment? The nitrogen treatment method needs to be supplemented.

2. In abstract, the authors mentioned that 15 BrNRT1 genes distinct to particular tissues, which may contribute to nitrate uptake and transport. However, the number of DEGs in the text exceeds 15, which is conflicts with abstract.

3. Many abbreviations, such as CLC, NPF, CRE, MFS, need to be presented in full at their first appearance.

4. There are 84 BrNRT1 genes within the Chinese cabbage genome. How many NRT1 genes in Arabidopsis, rice and tomato? What are the structural characteristics of the NRT1 genes?

5. Please renumber these 84 genes (BrNRT1.1-BrNRT1.84).

6. What does ‘motif organization’ mean in Figure 2?

7. The Figures of 1-3 are not clear and poor quality.

Comments on the Quality of English Language

1. The writing of this article is chaotic and the illogical. And there are also many language problems and grammatical errors. Please ask English personnel to revise.

Author Response

Dear Reviewers,

Thanks a lot for your useful comments and suggestions for our manuscript entitled “Genome-Wide Identification of the NRT1 Family Members and Their Expression under Low-Nitrate in Chinese Cabbage (Brassica rapa L. ssp. pekinensis)” (plants-2448395). We have modified the manuscript accordingly, and detailed corrections are listed below point by point.

Comment 1: Why choose the concentrations of 0.2 mM and 6 mM for nitrogen treatment? The nitrogen treatment method needs to be supplemented.

Response 1: Thanks for your suggestions. We have added the references on page **, line **. 

Comment 2: In abstract, the authors mentioned that 15 BrNRT1 genes distinct to particular tissues, which may contribute to nitrate uptake and transport. However, the number of DEGs in the text exceeds 15, which is conflicts with abstract.

Response 2: Thanks for your advice. We are very sorry for our negligence. We revised the manuscript in accordance with the reviewers’comments, and carefully proof-read the manuscript to minimize typographical, grammatical, and bibliographical errors.

Comment 3: Many abbreviations, such as CLC, NPF, CRE, MFS, need to be presented in full at their first appearance.

Response 3: Thanks for your advice. We have revised and unified the abbreviations throughout the text.

Comment 4: There are 84 BrNRT1 genes within the Chinese cabbage genome. How many NRT1 genes in Arabidopsis, rice and tomato? What are the structural characteristics of the NRT1 genes?

Response 4: Thanks for your suggestions. We have added a description of "the number of NRT1 genes in Arabidopsis, rice and tomato" in the text, on page **, line **.

Comment 5: Please renumber these 84 genes (BrNRT1.1-BrNRT1.84).

Response 5: Thanks for your suggestions. We have numbered these 84 genes according to the functional annotation of these genes based on the blast results of Arabidopsis, rice and tomato.

Comment 6: What does ‘motif organization’ mean in Figure 2?

Response 6: Thanks for your advice. We are very sorry for our negligence. We have corrected “motifs organization” to “conserved protein motifs organization” in the manuscript .

Comment 7: The Figures of 1-3 are not clear and poor quality.

Response 7: We have replaced the Figures.

Comment 8: The writing of this article is chaotic and the illogical. And there are also many language problems and grammatical errors. Please ask English personnel to revise.

Response 8: Thank you very much for the careful reading of our manuscript and valuable suggestions. We have made improvements to the text with English personnel. 

Round 2

Reviewer 1 Report

Comments and Suggestions for Authors

Table 1: Misspelled "log2". "2" should be written in subscript. The term 'Ratio' is unnecessary.
In order to simplify the visualization, the wrong analysis cannot be performed!!! The experiment was assumed to be multivariate and should be analyzed as such!!! This is unacceptable in scientific research. Such an approach disqualifies the manuscript to be considered for publication in any serious journal. The answer, on the other hand, I think is reprehensible!

Table S2 does not specify the number of degrees of freedom and F-values for the residuals. The title of this table is also incorrect.
Adding '**' markings is not an answer, and an addition, to the question: In what sense was Hidden Markov Model applied?
Still: The description of "6th Data Processing and Analysis" is very weak. It should be supplemented with all the statistical methods used.
Still: The Table with the results of the analysis of variance is not complete. The error information is missing. It should be corrected and completed.

Blatant errors, failure to correct the manuscript according to the comments, and absurd answers demonstrate the authors' lack of knowledge of the simplest assumptions and requirements of statistical analysis. The authors should consult a statistician, perform (correctly) the analyses and rewrite the manuscript.
In its current form, the manuscript is not suitable for publication.

Author Response

Dear Reviewers,

Thanks a lot for your useful comments and suggestions for our manuscript entitled “Genome-Wide Identification of the NRT1 Family Members and Their Expression under Low-Nitrate in Chinese Cabbage (Brassica rapa L. ssp. pekinensis)” (plants-2448395). We have modified the manuscript accordingly, and detailed corrections are listed below point by point.

Comment 1: Table 1: Misspelled "log2". "2" should be written in subscript. The term 'Ratio' is unnecessary.

Response 1: Thank you very much for the careful reading of our manuscript. We have made improvements to the "log2", and deleted 'Ratio'.

Changes: Page 7

Comment 2: In order to simplify the visualization, the wrong analysis cannot be performed!!! The experiment was assumed to be multivariate and should be analyzed as such!!! This is unacceptable in scientific research.

Response 2:Thank you for pointing out the serious errors in the article. We apologize for the mistake that occurred. We have revised Figure 4 to remove the icon for pairwise comparison of one-way ANOVA previously done. Table S2 has been added to supplement the results of the multivariate analysis of variance in detail. In addition, we have added statistical analysis content in “2.8. The expression profile of BrNRT1s in N Metabolism Pathway under LN Stress Conditions” and “3. Discussion”.

Changes: Page 9, Table S2

Comment 3: Table S2 does not specify the number of degrees of freedom and F-values for the residuals. The title of this table is also incorrect.

Response 3: Thanks for your advice. We have added the number of degrees of freedom and F-values for the residuals to the table S2, and also corrected the title of this table.

Changes: Supplementary File, Table S2

Comment 4: In what sense was Hidden Markov Model applied?

Response 4: Thanks for your suggestions. Statistical models called hidden Markov models are a recurring theme in computational biology. In this project, we used an online sequence analysis tool named HMMER to identify gene family members. HMMER is used for searching sequence databases for sequence homologs, and for making sequence alignments. It implements methods using probabilistic models called profile hidden Markov models (profile HMMs). HMMER is often used together with a profile database, such as Pfam or many of the databases that participate in Interpro. But HMMER can also work with query sequences, not just profiles, just like BLAST. This project meets the requirements for using HMMER. Xiunan Duan also used this method in his research paper "Identification and expression analysis of HMGR gene family in Glycine max"(Journal of Shanxi Agricultural University (Nature Science Edition), 2019, 39(4): 9-16).

Changes: Page 14

Comment 5: The description of "6th Data Processing and Analysis" is very weak.

Response 5: Thanks a lot for your careful reading of our manuscript. We have rewritten the part of "5. Materials and Methods".

Changes: Page 19

Again, we appreciate your insightful comment. We worked hard to be responsive to it. We hope our review meets your approval. Thank you for taking the time and energy to help us improve the paper.

Sincerely,

Li Jingjuan and Gao Jianwei

Reviewer 2 Report

Comments and Suggestions for Authors

The author has solved all my questions.

Author Response

Thank you so much for your comments.

Round 3

Reviewer 1 Report

Comments and Suggestions for Authors

The authors have incorporated all suggestions improve the manuscript. I recommend publishing it in its current form.